# The Role of Guilt and Empathy on Prosocial Behavior

**DOI:** 10.3390/bs12030064

**Published:** 2022-03-01

**Authors:** Costanza Scaffidi Abbate, Raffaella Misuraca, Michele Roccella, Lucia Parisi, Luigi Vetri, Silvana Miceli

**Affiliations:** 1Department of Psychology, Educational Science and Human Movement, University of Palermo, 90128 Palermo, Italy; michele.roccella@unipa.it (M.R.); lucia.parisi@unipa.it (L.P.); silvana.miceli56@unipa.it (S.M.); 2Department of Political Science and International Relations (DEMS), University of Palermo, 90134 Palermo, Italy; raffaella.misuraca@unipa.it; 3OASI, Research Institute-IRCCS, 94018 Troina, Italy; luigi.vetri@gmail.com

**Keywords:** guilt, empathic concern, prosocial behavior

## Abstract

Research on the effects of guilt on interpersonal relationships has shown that guilt frequently motivates prosocial behavior in dyadic social situations. When multiple persons are involved, however, this emotion can be disadvantageous for other people in the social environment. Two experiments were carried out to examine the effect of guilt and empathy on prosocial behavior in a context in which more than two people are involved. Experiment 1 investigates whether, in three-person situations, guilt motivates prosocial behavior with beneficial effects for the victim of one’s actions but disadvantageous effects for the third individual. Participants were faced with a social dilemma in which they could choose to take action that would benefit themselves, the victim, or the other individual. The findings show that guilt produces disadvantageous side effects for the third individual person present without negatively affecting the transgressor’s interest. In Experiment 2, participants were faced with a social dilemma in which they could act to benefit themselves, the victim, or a third person for whom they were induced to feel empathic concern. Again, the results show that guilt generates advantages for the victim but, in this case, at the expense of the transgressor and not at the expense of the third person, for whom they were induced to feel empathic concern. Therefore, guilt and empathy seem to limit the transgressor’s interest. The theoretical implications are discussed.

## 1. Introduction

For centuries, economists and psychologists have argued that moral emotions stimulate prosocial behavior. Prosocial behavior refers to the broad range of actions intended to benefit one or more people other than oneself [1] (p. 463). The field of prosociality is flourishing. However, researchers do not always agree on a common definition of prosocial behavior and often neglect to define it altogether. Common to almost all definitions is an emphasis on promoting well-being in agents other than the actor. For an overview of the breadth of definitions of prosocial behavior and the related concept of altruism and helping behavior, we refer the reader to the review by Pfattheicher et al. [2] in which the authors show how definitions of these concepts differ in whether they emphasize intentions and motivations, costs and benefits, and social context. The literature shows that guilt, as a moral emotion, does motivate prosocial behavior in dyadic social dilemma situations [3,4,5,6]. However, de Hooge et al. [7] claimed that in social situations involving multiple individuals the moral and prosocial nature of guilt is questionable. In these circumstances, indeed, guilt can produce positive consequences for the victim of one’s actions but disadvantageous effects for others in the social environment. For example, it might be considered moral behavior to spend more time with a hurt loved one at the expense of one’s own time. However, as de Hooge et al. [7] point out, while guilt may lead to an extra investment in that relationship, another person will have to pay the cost for this.

Building of de Hooge et al.’s [7] claim that in social situations involving multiple individuals the prosocial nature of guilt is questionable, in this paper we examine the effect of guilt in a social situation involving multiple individuals and we consider the beneficial role of empathy in inhibiting the negative effect of guilt. More specifically, in the first experiment carried out in the wake of that of de Hooge et al. [7] we investigate whether, in a three-person context, an individual’s guilt towards another person can give rise to prosocial behavior towards the victim in question but at the expense of a third person, who is not a victim, other than oneself. In the second experiment, we examine whether empathic emotions [8,9] towards the third person in the social environment might attenuate the highlighted effect (i.e., the negative impact on the third person from the transgressor’s prosocial behavior towards the victim). We will, therefore, turn our attention to empathic emotion, as a process that is activated when we stop focusing attention on our thoughts and emotions [10,11].

## 2. Interpersonal Effects of Guilt

The literature that addresses guilt attributes this feeling to an emotion characterized by a negative tone which is elicited when a person perceives that their behavior has violated moral standards or has caused harm to others [12,13,14]. As such, these are cases in which the actor has intentionally or unintentionally injured another person [15,16]. The agents are concerned about bad behavior and experience a lot of cognitive rumination [14,17]. Guilt has been described as a source of oppression and self-flagellation that may lead to unbearable feelings of self-loathing and despair as well as mental illness. Individuals high on guilt may feel worthless and deserving punishment [14]. Depression and guilt are both characterized by intropunitive traits and share common antecedents. Greater guilt is associated with greater depression and with symptoms of obsessive-compulsiveness, anxiety, somatization and psychoticism [18,19].

In the field of moral emotions, psychological theory and research distinguish between guilt and shame. Shame is an emotion typically experienced after failures, inadequacies and moral or social transgressions [20]. More specifically, it is experienced when people fail to live up to moral or social standards and when others are aware—or might be aware—of this failure. When people experience shame, they think of others who disapprove of them or who will evaluate them negatively. Shame is an emotion that leads to self-reflection: it is associated with an awareness of feeling small, worthless and incompetent [20,21]. Further, at the interpersonal level, shame is associated with the tendency to withdraw and isolate oneself [3,22].

In contrast to shame, guilt focuses on a specific behavior and does not generalize into a negative image of the whole self [16,23]. Tangney et al. [24] found that when referring shame-inducing situations, respondents were more concern with others’ evaluations of the self. In contrast, when describing guilt situations, respondents were more concerned with their effect on others. This difference in egocentric versus other-oriented concerns is not unexpected given that shame contains a focus on the self, while guilt relates to a specific behavior [25]. Differences can be identified in the neural basis of these emotions, as shame and guilt are related to activity in brain regions involved in social cognition and emotion regulation. However, they have distinct neural circuits that can be differentiated based on social evaluation [26].

Diametrically opposed theoretical positions and empirical findings are seen in work examining the role of guilt in prosocial behavior. Indeed, guilt has been theorized and empirically shown to be a negative feeling with many positive consequences [4,6,16,27,28,29]. For example, guilt theories assume that this emotion stimulates a better grasp of perspective and feelings of empathy [16,30]. There is also a great deal of the literature that suggests that guilt motivates people to make amends and redress their actions [3,24,31,32,33,34].

Many researchers demonstrated that individuals who were induced to feel guilty were more willing to help others, compared to other individuals who were not induced to feel guilty [35,36,37]. Guilt encourages behaviors that aim to restore the relationship between transgressor and victim or to prevent damage to this relationship [16,38,39]. In general, then, guilt is considered to be an exemplary moral emotion, motivating prosocial behavior and strengthening social bonds [3,40,41].

Economists have also addressed the guilt. Although they start from an egoistic view of the individual, they are also inclined to recognize that guilt has the advantage of holding back those personal egoistic tendencies and instead stimulating prosocial action [42,43,44]. For example, Ketelaar and Au [4] have shown that people who feel guilty after defecting in a social dilemma game tend to cooperate more in further repetitions of the game. Hopfensitz and Reuben [45] found that someone who is punished for defecting will only not retaliate in the future if they feel guilty or ashamed.

More generally, behavior following guilt is usually interpreted as moral behavior, or behavior motivated by concern for another person [35,46]. Therefore, guilt is often presented as a moral emotion that is associated with the well-being of society and that encourages people to think about how their own behavior affects the well-being of other people [3,6]

The most direct evidence of the moral effects of guilt can be found in recent studies concerning the effects of emotions on prosocial behavior in dyadic relationships. These studies demonstrate that people who feel guilty or anticipate guilt act in a prosocial way towards others when engaged in games involving social dilemmas [4,47,48,49,50]. Thus, in dyadic contexts, despite its negative tone, guilt plays a functional role in protecting interpersonal relationships, leading people to place others’ concerns before their own [3,29].

Although the motivational states generated by guilt that activate prosocial behavior are not the main focus of this work, it is important to note that the social psychology literature abounds of models explaining the psychological factors underlying the relationship between guilt and prosocial behavior. The main models that enjoy greatest empirical consideration are the following:The desire to repair the specific wrong [51]: A state of guilt causes the formation of a desire to repair the specific perceived wrong performed by the agent, which in turn motivates helpful behavior aimed at repairing the fault.The desire to repair wrong-doing as such [12,52]: This model posits that the motivation driving prosocial behavior is not so much a desire to right the specific wrong done but, rather, a more general desire to repair responsibility for a wrong as such.The desire to improve one’s own standing [53,54]: The previous model posited a desire, which is not directly concerned with the agent in question, but rather with morality itself and the importance of repairing a failure to live up to the agent’s moral standards. In order to help explain the relationship between guilt and helping, we could instead posit a potential desire on the part of the agent to improve their (actual or perceived) moral purity, worth, virtue, social image, social attachments, social and communal relationships, moral standing in the community, or the like.The desire to alleviate one’s guilt [12,55]. The fourth model to be mentioned here holds that guilt states often cause the formation of a desire to eliminate or reduce the agent’s guilt. Since helping is one very common way of making oneself feel better and no longer guilty about a prior wrong act, it is only to be expected that guilt would be positively correlated with helping, other things being equal. In this picture, then, helping is treated as an instrumental means for promoting the agent’s subjective well-being.

Beyond the factors underlying the relationship between guilt and prosocial behavior, guilt theories agree in seeing positive consequences for the well-being of others in feelings of guilt. This image of guilt is best summed up in the fact that a feeling of guilt is “an adaptive emotion, which benefits individuals and their relationships in a variety of ways” [40,56,57].

However, some authors have also indicated that this view of guilt may be too positive, and that a less prosocial side of guilt can be found. Therefore, they also focused on the negative interpersonal effects that guilt can stimulate, highlighting that the negative consequences are an integral part of the experience and function of guilt. In doing so, they turned their attention from a dyadic context to a context where the actor interacts with more people. Now, in a dyadic action, people worry about the damaged relationship and intend to repair the situation with the victim as soon as possible. Any action taken to improve the relationship with the victim can be interpreted as prosocial behavior because it highlights “helping another person at some sacrifice to oneself” [58] (p. 369). Indeed, empirical research reveals that when people with guilt are together with the victim only, they engage in behaviors that improve the victim’s outcomes at the expense of their own, such as giving money or buying gifts [4,44,47,48,50,59]. However, people are often not alone with the victim in their daily life. Rather, they often interact with multiple people at the same time. For example, after forgetting a sister’s birthday, people are likely to interact with other family members again. Moreover, after harming a colleague at work, people may continue talking and socialising with other colleagues and friends. Covarrubias and Fryberg [60], Zeelenberg and Breugelmans [39] and de Hooge et al. [7] investigated what happens when the person who feels guilty interacts not only with the person to whom they caused the damage but also with a third person. In this sense, these authors have examined the consequences of guilt in contexts that are not exclusively dyadic. The authors demonstrated how guilt drives people to enact prosocial behaviors towards those who generate feelings of guilt in them, but at the expense of other people present in the social environment. This potential harm to others calls into question the prosocial nature of this emotion when several people are involved. According to de Hooge et al. [7], the same judgments and inclinations provoked by feelings of guilt that benefit the victim can also lead to negative effects on other individuals present in the social context. When a person experiences guilt, their attention is temporarily focused on the victim’s pain, and consequently, the regard for the well-being of others is temporarily reduced. This means that when there is an opportunity to rectify the harm inflicted on the victim, this can happen at the expense of the other social partners [58]. Precisely because guilt prompts people to focus on their victims, it also causes temporary disregard towards others, with negative consequences for others’ well-being. For example, people may make peace with their sister by spending more time with her, time that is created by cancelling appointments with others. Alternatively, people may try to make amends with their deceived partner by putting more energy into that relationship and less energy into relationships with friends. For de Hooge et al. [7] this implies that a state of guilt does not necessarily evoke indifference to personal problems or alleviate the suffering of the victim at the transgressor’s own expense, but, rather, it evokes a sort of inattention towards the problems of others who are not victims. In sum, if we consider interactions involving more than two people, then the apparently moral behavior activated by guilt is effectively carried out at the expense of others, rather than one’s own detriment.

De Hooge et al. [7] tested this theory in situations where participants experiencing guilt interacted with two different partners—the victim and a non-victim—at the same time. The participants decided how to divide the resources between themselves, the victim and the non-victim, without the victim or non-victim having any influence on the division. In line with their reasoning, guilt motivated the participants to spend more resources on the victim but fewer resources on the non-victim compared to if they were in a neutral emotional state, and without renouncing to allocate resources to themselves.

How do the authors explain this result? Essentially, referring to the theories of fairness [61,62]. Equity theories such as the theory of interdependence [63,64] and the theory of social value orientation [65,66] claim that individuals are interested in both their outcomes and in the way these outcomes affect the well-being of other individuals in their social surroundings. In dyadic interactions, guilt temporarily yields a greater concern for the impact of one’s actions on the other one. In other words, when the interaction involves only two actors, guilt leads to negative consequences for the transgressor. However, when a third actor is involved in the interaction, the preoccupation for one’s interest is maintained along with the increased concern for the consequences produced in the victim. De Hooge et al.’s [7] conceptualization is in line with Walster et al.’s [67,68] work showing that when a transgressor experiences guilt, their behavior is motivated by the desire to restore equity and by the desire to maximize their outcomes, as well. The transgressor, thus, will try to repair the relationship by acting prosocially at the expense of the third individual in the social surroundings. It is important to specify that this does not mean that people intentionally hurt a third party, but rather that their attention on the relationship with the victim may cause them to involuntarily neglect the impact of their behavior on others’ well-being [7]. Consequently, victim-oriented motivation directs the transgressor’s attention to the victim themselves, thus translating it into restorative and altruistic behavior but without, on balance, the transgressor having to pay for it.

## 3. Introducing an Altruistic Motive

When will an individual act to benefit a victim they have harmed while avoiding harming a third person purely to pursue their own self-interest? In our opinion, one situation in which this might occur is when the transgressor feels a simultaneous motivation both toward the victim and the third person. In a three-person context, the motivation determined by the guilt benefits the victim but at the expense of a third person, because the transgressor is unlikely to give up favoring themselves as well. When this is contrasted with an empathic drive towards a third person, the transgressor may renounce their own satisfaction by maintaining the restorative action caused by the guilt.

Evidence from more than 30 experiments supports the empathy–altruism hypothesis, the hypothesis that empathic concern produces altruistic motivation [11,69,70,71]. Empathic concern has been conceptualized as the main source of altruistic motivation. Batson defined empathic concern, as an other-oriented emotional response generated by, and congruent with, the perceived welfare of a person in need. Empathic feelings include sympathy, compassion, tenderness and the like. Empathic concern is other-oriented in that it involves feelings for the other. This concern is not only about perceiving the other as needy, but also taking the other’s perspective, imagining how the other may have been influenced by their situation [10,11,72].

If empathic concern leads to altruistic motivation, then valuing the welfare of and feeling for another person in a particular situation should introduce an entirely new motive to a social dilemma (alongside guilt reduction and personal self-interest): a motive to benefit that person. In short, the empathically concerned individual should feel pulled, not in two directions but three. Empathic concern should activate a real interest in the other (in this specific case, towards the non-victim third person) and as such to the detriment of their own personal interest. If this is so, the transgressor should in that case show both a drive towards altruistic behavior in relation to the victim they feel guilt over and a drive towards altruistic behavior in relation to the third person for whom they feel empathic activation.

To test this hypothesis, we ran two experiments that focus on the effects of guilt (Experiment 1) and the effect of guilt and empathy (Experiment 2) toward a third (non-victim) person present in the experimental setting.

## 4. Experiment 1

Experiment 1 investigates whether guilt motivates prosocial behavior in three-person situations in a social dilemma, with beneficial effects for the victim as well as disadvantageous effects for a third, non-victim individual present in the social environment.

### 4.1. Hypothesis

As in de Hooge et al.’s experiment [7] we test the hypothesis that guilt generates prosocial behavior in three-person situations but creates disadvantages for others. In particular, the transgressor will act to the benefit of their victim but to the detriment of a third person in order to maximize their own profit.

### 4.2. Method

#### 4.2.1. Participants and Design

Data were collected from a convenience sample of undergraduate students enrolled at a university in Italy (Palermo). Thirty university students (20 women, 10 men; M *age* = 22.14 years, SD = 2.15) participated in the experiment. Participants received course credit for their participation We used a student sample in experiments 1 and 2 not only because of their homogeneity such as age and education, but also because we wanted to reproduce the same conditions of de Hooge et al.’s study employing a student sample. We recognize that our investigation possesses the characteristics of a limited laboratory test that cannot generalize to other samples. They were randomly assigned to one of the two conditions, guilt or control (15 in each group). The experiment consisted of three tasks: a guilt-induction procedure, a test to assess the emotion measures and a three-person dictator game that assessed lottery ticket allocation to the victim, to the third person and o the self. The study was approved by the Research Ethics Board of Palermo University.

#### 4.2.2. Procedure and Materials

Participants were told that they would be engaged in a task named “Letter Task” in which they could earn lottery tickets. The tasks where participants were engaged were, actually, the following three:

*First task: Guilt Manipulation*. During the first task, each participant was paired ostensibly with one other participant. They were sitting in front of each other and were in front of a computer screen to execute two rounds of a performance task (adopted from Reitsma-Van Rooijen et al. [73]). For this task, we used a guilt induction procedure that was used by De Hooge et al. [7]. Letters appeared rapidly on the screen in either red or green. In order to earn points, participants had to promptly respond to green letters by pressing the corresponding letters on the keyboard before they disappeared from the computer screen. The other player could earn points in a similar way for the red letters. After 3 min, their total scores were calculated and feedback was given.

Participants were informed that in the first round they were performing the task to win the lottery tickets for themselves, while in the second round they were performing the task to win tickets for the other player. Both players were required to perform well enough to reach a minimum level of 100 points to get three lottery tickets. After the first round, all participants received false feedback.According to APA (APA Style 7th Edition, 2019) guidelines, participants’ deception about the study’s true purpose is considered appropriate only when the research hypothesis could not be tested in any other way and the scientific knowledge gained from the study outweighs its costs. This was the case in our study. Consequently, we provided false feedback to the participants as participants might not have behaved naturally without it that they and their partners had performed well enough to earn tickets. Participants then performed the second round of the “Letter Task” to win the tickets for their partners. Participants assigned to the guilt condition were told that their performance was not good enough for the other player to receive three lottery tickets. Participants assigned to the control condition were told that the other player would not receive tickets because their own performance was not good enough. In other words, the results in the guilt and in the control condition were the same, but who was responsible for these results has changed.Consequently, in this and Experiment 2, the control group, actually, can be seen as an intervention group. 

*Second task: Emotion Measures.* Participants indicated on a scale from 1 (*not at all*) to 7 (*very much*) to what extent they felt guilty for their actions, how much they regretted their effort and how responsible they felt. These three questions assess the basic elements of guilt [74,75]. As in de Hooge et al. [7], to verify that the observed effects were unique to feelings of guilt and not driven by other negative feelings, we also assessed to what extent participants experienced shame, a closely related but different emotion. Participants indicated to what extent they felt ashamed, embarrassed and bad about themselves.

*Third task: Ticket Division.* At the end of the second task, participants played a three-person dictator game. The dictator game included the participant (whom we will henceforth refer to as the “transgressor”), the player who was ostensibly paired in the earlier “Letter Task” with the transgressor (whom we will henceforth refer to as the “victim”) and the “third player” who had not performed the earlier “Letter Task”. The second and third players were two confederates of the experimenter, and they were always the same persons across the tasks.

Participants were informed that they would receive either three or six lottery tickets that were left over from the first task and that their task was to allocate them among the three of them. Participants were always given six tickets and were told that the other two players did not know how many tickets they had received [76]. Since the other two players were not aware of how many tickets the participant held, they could potentially divide the tickets unequally without appearing unjust. The numbers of tickets participants allocated to the victim, the third player and themselves were our dependent measures.

Once participants had completed this measure, they answered questions about what they believed the purpose of the study to be. No participants indicated suspicion concerning the real aim of the study. Following this, they were given a full explanation of the experiment.

### 4.3. Results

#### 4.3.1. Emotion Manipulation Check

Participants assigned to the guilt group scored higher on all guilt variables than did control participants, all *ts*(28) > 4.57, *p* = 0.002,. Guilt participants reported more guilt (M = 5.60, SD = 1.99) than did control participants (M = 2.73, SD = 1.39), *t*(28) = 4.57, *p* < 0.001, They also reported more guilt than other emotions, all *ts*(14) > 7.56, *ps* < 0.001.

#### 4.3.2. Ticket Division

The results of the experiment are shown in Table 1. Guilt participants allocated more tickets to victims than did control participants, *t*(28) = 3.44, *p* = 0.002. In addition, they allocated less tickets to the third player than did control participants, *t*(28) = −2.65, *p* = 0.013. Guilt participants did not differ statistically from control participants in the amount they kept for themselves, *t*(28) = 1.18, *p* = 0.24.

Guilt-motivated participants to give more tickets to the victim, but fewer tickets to the non-victim compared to if they were in a neutral emotional state. Interestingly, participants experiencing guilt do not differ from participants in neutral states in terms of the amount of tickets they keep for themselves.

### 4.4. Discussion

In Experiment 1, we replicated the findings of de Hooge et al. [7] showing that guilt motivates prosocial behavior toward the victim with negative consequences for other social partners. In the specific case of guilt, the transgressor’s actions benefited the victim to the detriment of a third person to preserve benefit to themselves. Guilt theories certainly suggest that the function of guilt is to protect and improve social relationships in general, which would imply that this emotion should have positive consequences for everyone in the agent’s environment [4,6,12,29]. However, this may not always be the case. Thus, if we define prosocial behavior as helping others at a cost to the self [77,78], then this potential harm to others calls into question the prosocial nature of this emotion when several people are involved. This result is not surprising. In a world where there are not sufficient resources to go around, the power of self-interest is all too evident [79,80]. The social sciences generally take a pessimistic view, arguing that in situations such as the dictator game, individuals will prioritize their own interests unless measures are taken to prevent it [8,81]. Human beings’ fundamental instinct is to act in self-interest, which governs most of their interactions with one another [82,83,84]. The literature that explores the relationship between guilt and prosocial behavior highlights the idea that when an individual experiences guilt, they momentarily experience elevated interest in the outcome of the other person. What psychological factors account for this relationship or, more precisely, what motivational states guilt gives rise to, which, in turn, foster prosocial behavior, are not, however, the subject of this research. Frequently, in dyadic situations, this prosocial behavior comes at the expense of the individual’s personal outcome [4,6,12,29,40,44,50]. In the current experiment, in which a third party was present, the usual concern for one’s personal outcome was observed along with a heightened, guilt-induced concern for the outcome of the victim. This result could basically be said to be in line with equity theory, which focuses on determining whether the distribution of resources is fair to both relational partners [85,86]. As suggested by Walster et al. [68] (p. 190), when experiencing guilt “the harm doer is not only motivated by a desire for equity restoration, but also will act in such a way as to achieve the highest possible profit and satisfaction”. What we have confirmed is that the restoration of equity through remedial behavior in favor of the victim and the maximization of personal profit can only coexist by being disinterested in the third person present in the social context.

## 5. Experiment 2

In Experiment 1, the conflict between the distribution of resources between oneself, the victim and a third person was resolved to the disadvantage of the third person. As in de Hooge et al. [7], after the experience of guilt, participants kept their focus shared between self-interest and interest in the victim.

Experiment 2 examines what happens in the participant’s allocation of resources between themselves, the victim and the third person—whether the transgressor feels two parallel motivational drives this time, one toward the victim arising from guilt and the other toward the third person arising from empathic concern. We believe that when empathic concern for a third person is introduced, the resulting altruistic motivation will create additional conflict. The conflict is between responses the participant feels are all necessary: guilt generates a need for the participant to repair the damaged relationship because of the harm caused to the victim; empathic concern for the third person induces the participant to behave altruistically towards them as well; and, finally, the subject wants to maximize their own profit. Who succumbs?

We predicted that when a transgressor feels empathic interest in a third person who is not a victim, any allocation of resources to the target of empathic concern would likely be at the expense of self, without reducing the benefit to the victim. In Experiment 1, we had two motives in play, self-interest and guilt-reduction; in Experiment 2, three motives were in play, self-interest, guilt-reduction and empathy-induced altruism. We hypothesized that the latter two would be strong enough to limit self-interest.

### 5.1. Hypothesis

We expected that transgressors would allocate more lottery tickets to the source of their guilt, the victim, but at their own expense rather than at the expense of the third person for whom they felt empathy (altruism condition). This would contrast with the findings of the previous study, in which the transgressors allocated more lottery tickets to the victim at the expense of the third person for whom empathy had not been induced (baseline condition).

### 5.2. Methods

#### 5.2.1. Participants and Design

Data were collected from a convenience sample of undergraduate students enrolled at a university in Italy (Palermo). Sixty university students (40 women, 20 men; M = 23.19 years, SD = 2.10) participated in the experiment. Again, participants were recruited from undergraduate psychology courses through adverts on the webpage of the social psychology professor leading the research. Participants received one course credit for their participation. Fifteen subjects in each group were randomly assigned to the conditions of a 2 (emotion: guilt vs. control) × 2 (empathy: no communication vs. communication–empathy) design. The study was approved by the Research Ethics Board of Palermo University.

#### 5.2.2. Procedure and Materials

The procedure of this study is the same as the experimental procedure used in Experiment 1 except for the addition of the manipulation and the measure of the empathy variable.

*First task: Guilt Manipulation* Participants entered in the lab and performed the same “Letter Task” as in Experiment 1.

*Second task: Emotion Measures* Participants answered the same emotion-manipulation assessment questions as in Experiment 1.

*Third task: Ticket Division* At the end of the session, participants played the three-person dictator game as in Experiment 1, along with their partner from the “Letter Task” (victim) and a person named Antonella, who had not performed the earlier “Letter Task” (third player). Thus, as in Experiment 1, the second and third player were two confederates of the experimenter, and they were always the same persons across the tasks.

*Empathy Manipulation: Using Perspective to Induce Empathy.* Participants were randomly assigned to one of two experimental conditions. Participants assigned to the no-communication condition made their allocation decision in the absence of information about the other two players (the victim and the third player). Those assigned to the communication–empathy condition received a note that had ostensibly been written by the third player immediately on arrival for the experiment, before learning anything about the nature of the experiment. All participants in the communication–empathy condition received the same hand-written note signed “Antonella.” The note described being down after having recently been dumped by a long-term boyfriend. Participants in the communication–empathy group were instructed to imagine how the note writer felt with respect to what she had written about in the note. This empathy-induction technique has been widely used in experimental research on this emotion [8,58].

*Empathy Measures.* After reading the note and reflecting on it for a short time, participants completed an impressions and feelings questionnaire. In this questionnaire, they used a 7-point scale (1 = *not at all*, 7 = *extremely*) to indicate to what degree each of a list of emotion adjectives described how they were feeling toward Antonella. Included among the adjectives were six that had been used in previous research to measure empathy [8] (see Batson, 1987): sympathetic; warm; compassionate; soft-hearted; tender; and moved. These self-reports were used to assess the effectiveness of the empathy manipulation.

### 5.3. Results

#### 5.3.1. Empathy Manipulation Check

Participants indicated in the final questionnaire the extent to which they remained objective and the extent to which they imagined the feelings of the person who had written the note (1 = *not at all*, 9 = *totally*). The objectivity score was subtracted from the imagination score to create an index of the perspectives adopted (empathic objective). This difference measure revealed much higher scores for imagining relative to objectivity in the altruism condition as compared to the baseline condition (M = 1.67 SD = 1.02 and M = 2.46, SD = 1.38 respectively), *t*(58) = 13.10, *p* < 0.001.

To evaluate the emotional response of empathy, we created an index combining the scores for the six adjectives included in the empathy measure. The index presented adequate consistency (α = 0.84). Participants in the communication–empathy condition reported greater empathy (M = 4.96, SD = 0.88) than those in the no-communication condition (M = 3.31, SD = 0.84), *t*(58) = 7.37, *p* < 0.001.

#### 5.3.2. Emotion Manipulation

Guilt participants scored higher on all guilt variables than did control participants, all *ts*(58) > 4.88, *ps < 0*.001. Guilt participants reported more guilt (M = 5.23, SD = 1.89) than control participants (M = 2.63, SD = 1.27), *t*(58) = 6.30, *p* = < 0.001. They also reported more guilt than other emotions, all *ts*(29) > 8.61, *ps* < 0.001.

#### 5.3.3. Ticket Division

Table 2 summarizes the means of tickets allocated to self, to the victim and to the third person (Antonella) in each experimental condition. In the no-communication condition, the pattern of results resembles those of Study 1.

We carried out three ANOVAs to analyse the effects of empathy and guilt on the three distribution decisions: tickets for oneself, for the victim and for the third person.

*Dependent Variable: Tickets for oneself.* A 2 (emotion condition: guilt vs. control) × 2 (communication condition: empathy vs. no communication) analysis of variance with tickets allocated to oneself showed significant main effects of the empathy condition, *F*(1,56) = 38.99, *p* ≤ 0.001, η^2^ = 0.38 and of guilt *F*(1,56) = 3.75, *p* = 0.058, η^2^ = 0.037. The mains effects were qualified by an interaction guilt X empathy *F*(1,56) = 3.70, *p* = 0.05, η^2^ = 0.036. Post-hoc pairwise comparison with Bonferroni correction revealed that participants in the communication–empathy condition offered significantly fewer tickets to themselves in the guilt condition than they did in the control condition, *t*(56) = −3.04 *p* = 0.004. By contrast, there was no difference between the guilt and control condition with regard to participants in the no-communication condition *t*(56) = 3.55e, *p* = 0.98 (see Figure 1).

*Dependent Variable: Tickets assigned to the victim.* A 2 (emotion condition: guilt vs. control) × 2 (communication condition: empathy vs.no-communication) analysis of variance with tickets allocated to the victim showed only a main effect of the guilt condition in the number of tickets for the victim, *F*(1, 56) = 14.75, *p* < 0.001, η^2^ = 0.20. Participants in the guilt condition assigned more tickets to the victim compared to those in the control condition, *t*(56) = 3.84, *p* < 0.001.

*Dependent Variable: Tickets assigned to the third person (Antonella).* A 2 (emotion condition: guilt vs. control) × 2 (communication condition: empathy vs.no-communication) analysis of variance with tickets allocated to the third person revealed a main effect of the empathy condition, *F*(1,56) = 143.83, *p* < 0.001. η^2^ = 0.67. The main effect was qualified by an interaction guilt x empathy, *F*(1,56) = 10.70, *p* = 0.002, η^2^ = 0.05. Post-hoc comparison with Bonferroni correction showed that participants offered less money to the third player when they were in no-communication condition, *t*(56) = −3.60, *p* = 0.004. By contrast, there was no difference between guilt and control condition concerning participants in the empathy communication condition *t*(56) = 1.10, *p* = 0.73 (see Figure 2). The third person received fewer lottery tickets in the guilt condition than in the control condition when the empathy manipulation occurred.

### 5.4. Discussion

The results of this second experiment highlight two main aspects. First, the motivation that leads to the reduction of guilt is maintained. This can be inferred from the behavior of the transgressor who allocates more tickets to the victim in the guilt condition, compared to the no-guilt condition. Second, the empathy induced towards the third party leads to an altruistic behavior. The transgressor, indeed, assigns more lottery tickets to the person for whom they are induced to feel empathy. As a consequence of the above results, the maximization of one’s own personal profit diminishes.

## 6. General Discussion

Guilt is a social emotion that is strongly linked to interpersonal behaviors towards others in one’s environment. Prior research has shown that guilt motivates interpersonal behavior with positive consequences and empirical research on guilt in dyadic situations supports this view [4,44,47,48,50,59]. Yet, de Hooge et al.’s research [7,59] reveals that the most important effect of guilt may not be the positive influence on certain relationships. Guilt can lead to such concern for repairing the damage done to the victim that the well-being of others in the same social environment is forgotten. Consequently, transgressors may be motivated to repair the damage at the expense of another. In the first experiment, this effect was confirmed: guilt motivates prosocial behavior, but to the detriment of a third person and not at the transgressor’s own expense. Thus, guilt may motivate behavior that does not fit perfectly with its conception as a moral emotion.

The discovery that guilt can also produce adverse side effects for non-victimized others makes the interest of the research clear, in our view. Indeed, the main significance of our research lies in the observation that moral emotions do not cause people to indiscriminately neglect their personal interest. In other words, it could be said that the behavioral consequences of guilt can be more or less moralistic depending on the motivational functions that are triggered.

In the second experiment, individual interest, motivation to repair the guilt and empathic concern for others are the variables considered to understand how individuals interact on the basis of the constraints that limit their actions and the goals that drive them to act. Empathic concern for the well-being of others and the motivation to repair a sense of guilt pushes us to take into account the effect that our actions have on others and can thus contribute to achieving socially desirable results. It is possible to imagine these motivations actually coming together in real life. For example, in the workplace we may find ourselves in a situation where we both feel a sense of guilt towards a colleague who has been harmed by us (intentionally or unintentionally) and an empathic activation towards another colleague. If the context requires it, both motivations could turn out to be a kind of pressure to carry out unselfish behavior towards both colleagues, not overlooking one or the other but instead putting aside our own interest.

The fact that the second experiment showed that guilt seems to be a basic need of the individual raises several considerations that point to the significance of the study. The first consideration that we believe is appropriate to make could be related directly to what was stated earlier in reference to the results of the first experiment. In that case, we emphasized how potential harm to others can call into question the prosocial nature of this emotion when multiple people are involved. Now, what we note in the second experiment is that the motivation that leads to the reduction of guilt is still maintained. In fact, when another motive (in our case, empathy-induced altruism) is added to self-interest and to guilt reduction, a transgressor does not renounce to benefit the victim towards whom they feel guilty, nor do they renounce to benefit a third person for whom they feel empathy. The transgressor, instead, becomes more willing to renounce to their own interest [41,87]. In terms of the effects of guilt, our primary contention is that reducing feelings of guilt is a fundamental necessity for the individual, otherwise they would not abandon their own interest in favor of the victim. Whether the real motivation at the basis of the guilt/social behavior relationship is selfish or altruistic, guilt still has a great deal of power over the individual. Secondly, in terms of empathy, the second experiment demonstrated that empathy is a “threat” to self-interest, not to the interest of the victim, to whom the individual continued to distribute resources.

The second consideration leads us to results presented in the literature related to the effects of empathy in social dilemmas. Classical game theory is one of the approaches that has paid the most attention to social dilemma [88,89,90]. The cornerstone of this theory is the assumption that human beings generally act out of self-interest, so most of an individual’s interactions with others is governed by self-interest. However, classical game theory acknowledges that the individual can also act for the collective good. In a range of circumstances, a substantial proportion of resources are allocated to the benefit of the group at a cost to the self. People donate money to public television and radio; they recycle even when it is inconvenient; and they donate during blood drives with no strings attached [91,92]. Batson and colleagues [93] point out a third element. In addition to benefiting oneself as an individual or benefiting the collective, one may act to benefit another individual as an individual to the detriment of the group as a whole. Based on research on altruism, Batson [8,93] raises the possibility that empathy towards another person induces an altruistic motive that can have paradoxical consequences. If, in a social dilemma, an individual feels empathy toward another individual in the group, their desire to increase the well-being of the object of that empathy may lead them to act to benefit that individual, consequently reducing the resources available for the group as a whole. Thus, the inclusion of this third element may represent a new threat to the collective good. Several experiments, for example, suggested that, in a multiple-player game involving social dilemmas, empathy increased prosocial behavior towards an individual, but the behavior came at the expense of the collective good without affecting the allocation of resources to the individual themself [93,94]. Indeed, in certain nontrivial circumstances, empathy can pose a more powerful threat to the collective interest than self-interested egoism. We did not investigate, as Batson and Ahmad [93], whether empathic concern for a person occurs at the general expense of the collective good but rather whether it occurs at the expense of one’s own interest or at the expense of one’s victim. In our second experiment, prosocial behavior motivated by empathy resulted in greater personal expense than prosocial behavior motivated by guilt. While Batson and Ahamd [93] suggested empathy may be a threat to the collective good, our research shows empathy is more of a threat to self-interest.

Furthermore, it is worth noting the value of this research lies also in its highlighting the often neglected role of guilt and empathy in contexts that involve multiple persons. Research on the consequences of guilt and empathy has focused primarily on dyadic relationships. In real life, however, our responses to our own emotions change depending on whether the context is one of a dyadic relationship or one in which a third person is involved. The only research that investigated the consequences of guilt in contexts that were not exclusively dyadic are the previously mentioned studies by Zeelenberg and Breugelmans [39] and de Hooge et al. [7]. Similarly, research in the literature mainly investigated the effects of empathy within dual relationships, meaning the effects on the person for whom one feels empathy. However, beyond the direct effects about our empathy, what are the consequences for the other people present in the social context? Does empathy refocus all the individual’s prosocial behavior towards the subject of that empathy, and, therefore, does this same individual no longer act selflessly towards the subject of their guilt? Based on the results of our second experiment, empathy does not decrease a person’s altruistic behavior. Instead, the individual maintains a substantial focus and attention on their victim as well.

We would like to make two closing notes concerning the de Hooge paradigm [7] we used in our studies. First, it could be argued that the de Hooge paradigm is not ideal for studying guilt because there is not a clear transgression. As the literature on the subject tells us, guilt is primarily characterized by the person’s idea of having done a bad thing or having failed at some specific thing, with a feeling of remorse/regret for the wrong thing they have done. Usually, people experience guilt when they feel responsible for damage to a relationship with another person [12]. The central signal of guilt thus concerns the negative impact of people’s actions on their relationship with a specific other (the victim). If we follow the interpersonal approach, according to which “by guilt we refer to an individual’s unpleasant emotional state associated with possible objections to his actions and based on the possibility that one may have done something wrong” [12] (p. 245), then we believe that the manipulation of guilt in our experiments may be suitable. In fact, the manipulation requires the participant to be told that they have performed incorrectly, and for that reason their partner will not receive any lottery tickets. Therefore, the participant actually did something wrong: they harmed a relationship partner and received benefits (the lottery tickets), while the harmed partner received none. “The knowledge that one has harmed another person may be enough to cause guilt” [12] (p. 245). Furthermore, proneness to guilt may become generalized to other relationships, including even minimal intergroup phenomena [95]. In our view, the De Hooge paradigm [7] makes the subject focus on their own bad performance and worry about the effects of that specific behavior on others [40].

Secondly, it could be said that the de Hooge paradigm does not demonstrate neglect to a third party since the smaller allocation of tickets would not represent a cost to third party because they had no reason to expect any. We are not sure about this logic. It is true that with regard to the distribution of lottery tickets, the third person expects nothing and, therefore, suffers no harm or cost, but it will certainly be an unfair distribution on the part of the transgressor and we believe being subjected to an unfair distribution is always harmful [96,97].

This study has the main limitation of a small sample size, which might have decreased statistical power and increased the margin of error. Further, the use of convenience samples of undergraduate college students as subjects in our investigation leads to a consideration. College student subjects might enhance research validity because of their homogeneity; such apparent homogeneity makes this sample easier to compare than other group of people because of their demographic and psychographic characteristics [98]. However, certain personality characteristics of college students may have been crucial elements in these experiments in which we investigated the effects of empathy. We may be questioned, for example, whether the empathy manipulation would have been successful with a different type of subjects. Thus, replications with larger and more representative samples are needed in order to provide a better understanding of the impact of guilt and empathy on prosocial behavior.

Further studies are also needed in order to answer some of the questions left unanswered by our research. For example, we wonder if the effect would be observed with two (or more) non-victims. Since the subject has been induced to empathize with a specific third person who is not a victim, the question to be asked is whether it is possible for the empathic focus to be directed towards two or more people at the same time. Furthermore, we considered the division of the lottery tickets. It would be interesting to investigate whether the same results would occur with other variables, for example, charitable donations or money, or with a less restricted resource, such as time or working hours [99]. An extension of our results to less evidently limited resources would support the strength and generalizability of the effect.

Prosocial behavior is a broad and multifaceted concept, which refers to different types behaviors (e.g., altruism and cooperative behavior). Although these different behaviors are correlated with each other, their links may change overtime, since they are influenced by the age of the individuals, and by cognitive and situational variables. Nevertheless, the capacity to help and support other individuals is certainly one of the ideal aims to achieve. Among the urgent needs of the contemporary society, the need to enhance moral qualities in order to get out of self-centeredness seems to be one of the priorities unanimously felt. For this reason, the psychological research on pro-social behavior and in particular on the antecedents and correlates of phenomena such as altruism and cooperative behavior seems to be of great importance for the educational domain [100,101].

## Figures and Tables

**Figure 1 behavsci-12-00064-f001:**
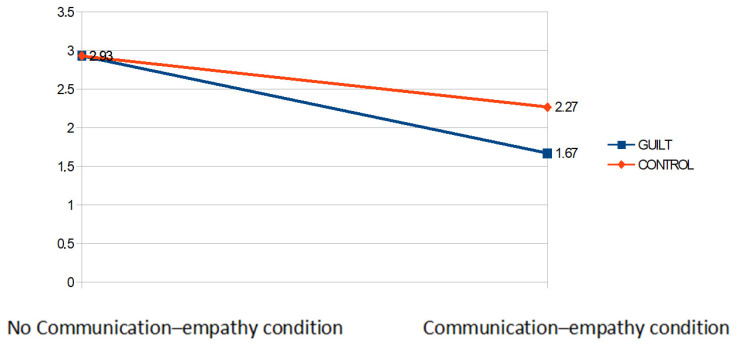
Mean of the tickets allocated to self in each experimental condition.

**Figure 2 behavsci-12-00064-f002:**
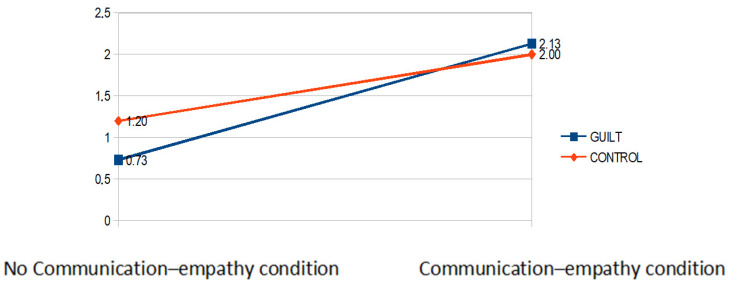
Mean of the tickets allocated to third person in each experimental condition.

**Table 1 behavsci-12-00064-t001:** Means and standard deviations of the lottery tickets distribution among the three individuals in each condition of Experiment 1.

	Guilt		Control	
	M	SD	M	SD
Transgressor	2.46	0.51	2.73	0.70
Victim	2.73	0.45	2.06	0.59
Third Person	0.80	0.41	1.20	0.41

**Table 2 behavsci-12-00064-t002:** Means and standard deviations of the lottery tickets distribution among the three individuals in each condition of Experiment 2.

	Guilt		Control	
**Empathy-Communication**	M	SD	M	SD
Transgressor	1.67	0.48	2.27	0.45
Victim	2.20	0.41	1.73	0.45
Third Person	2.13	0.35	2.00	0.40
**No-Communication**	M	SD	M	SD
Transgressor	2.93	0.79	2.93	0.59
Victim	2.33	0.61	1.86	0.35
Third Person	0.73	0.45	1.20	0.41

## Data Availability

Data supporting the results of this study are available at the link https://drive.google.com/drive/folders/1bL2mC43xWTejKc7T5BK4-PtaYip-bWVr?usp=sharing. (Accessed on 12 September 2021).

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
