# Peer review of "The Role of Guilt and Empathy on Prosocial Behavior"

_behavsci, 2022, doi:10.3390/bs12030064_

Round 1

Reviewer 1 Report

Although the objective of the research is interesting, the sample used does not justify its publication in the journal. This should be more extensive and the choice of university students justified.

I recommend that the authors replicate this study with a larger sample and with other characteristics.

Author Response

Point 1: Although the objective of the research is interesting, the sample used does not justify its publication in the journal. This should be more extensive, and the choice of university students justified.

Response 1: We fully understand the problem related to the small sample size. The small sample size may presumably have been a limiting factor in finding significance to the customary .05 level. In the Limitations we have highlighted this aspect of the study. We tried to look at the statistical power of the analyses, as this was not research based on random and representative sampling of the population. Our view was that under the conditions of the study, the power was adequate for the numerosity used. We considered Cohen's table referring to the t-test, which compares numerosity based on probability level, expected power, and effect size.

  • Kraemer, H. C., & Thiemann, S. (1987). How many subjects? Statistical power analysis in research.Sage Publications, Inc.

It is also true that many other published papers in this field, which used behavioral measures as dependent variables, had a similar sample size; see for example:

-           Basil, D.Z., Ridgway, N.M. and Basil, M.D. (2006), Guilt appeals: The mediating effect of responsibility. Psychology & Marketing, 23 (Study 2)
-           De Hooge ( 2013) in C. Mohiyeddini, M. Eysenck, & S. Bauer (Eds.), Psychology of Emotions. Nova Publishers (Study 1 and 2)
-           De Hooge, I. E., Nelissen, R. M. A., Breugelmans, S. M., & Zeelenberg, M. (2011). What is moral about guilt? Acting “prosocially” at the disadvantage of others. Journal of Personality and Social Psychology, 100(3), 462- 473. (Study 4).

-           Amoh, N. (2021). Youth Bystander Intentions to Intervene in Peer Intimate Partner Violence: The Co-influence of Perceived Perpetrator Race and Perceived Culpability (Doctoral dissertation, City University of New York).

Reviewer 2 Report

In this article the authors present an investigation of the effects of guilt in interpersonal relationships and its effect on prosocial behaviour in dyadic social situations when several people are involved. In the first experiment, in which participants were faced with a social dilemma in which they could choose an action that would benefit themselves, the victim or another individual, the results showed that guilt produces disadvantageous side effects for the third individual present without negatively affecting the interest of the transgressor. In the second experiment in which participants were faced with a social dilemma in which they could act to benefit themselves, the victim or a third person for whom they were induced to feel empathic concern, the results showed that guilt generates advantages for the victim at the expense of the offender and not at the expense of the non-victim third person. Therefore, in this paper, guilt and empathy seem to limit the transgressor's interest.

The article is well written and easy to read. However, there are some details that are not entirely clear in the article.

  • Regarding the payment that participants receive, it is not clear whether there is a fixed payment for participation or the tickets or points that the most and the least get, the distribution. This is important as it may motivate behaviour.
  • Although I am not in favor of giving false information to the participants, I understand the reason for doing it, although its justification should be explained a little more in the article.
  • One of the things that should be emphasized in the article is that in the experiments the control group is an intervention group. For example, when they are told that the other does not get tickets because of their own behavior or because of the player's behavior, they are intervening in both situations, I would have liked to know the result of round two if they had not been given the information above and compare it with those two situations.
  • On the other hand, I have some doubts about the interest of this type of work in the academic literature. The bibliography presented is very old, only 10.8% of the 74 articles have been published between 2011 and 2022, while only 2.7% in the last 5 years. Is there no more recent bibliography? Either the authors have not done a correct bibliographic search, or it is a topic that is no longer of much interest. Although I am not willing to accept the latter, since all the prosocial situations and the effect that empathy and other types of relationships and emotions have, are on the rise in the decision-making literature.

Author Response

Point 1: Regarding the payment that participants receive, it is not clear whether there is a fixed payment for participation or the tickets or points that the most and the least get, the distribution. This is important as it may motivate behaviour.

Response 1: We reported on page 6 "Participants received course credit for their participation".

Participants were told that in addition to course credits, they could also earn lottery tickets. The section of the procedure has been made clearer on page 6.

Point 2: Although I am not in favor of giving false information to the participants, I understand the reason for doing it, although its justification should be explained a little more in the article.

Response 2: As it is stated in the manuscript, we adopted de Hooge et al.’s (2011) procedure, which includes some deceptive methodologies. We now added a brief justification for the use of false feedback (p.6 footnote 2). Deception is was necessary for the purpose of our study because without the provision of false information, we could not have manipulated the considered independent variable.

Point 3: One of the things that should be emphasized in the article is that in the experiments the control group is an intervention group. For example, when they are told that the other does not get tickets because of their own behavior or because of the player's behavior, they are intervening in both situations, I would have liked to know the result of round two if they had not been given the information above and compare it with those two situations.

Response 3: The control group (with respect to the manipulation of the guilt) receives the above information because it is necessary for it to know that there is a person in the second task who did not get anything (although in this case it is his/her responsibility and not the responsibility of the participant to whom he was paired in the first experiment). Certainly in this regard is also an intervention group. We reported this on p.6

Point 4: On the other hand, I have some doubts about the interest of this type of work in the academic literature. The bibliography presented is very old, only 10.8% of the 74 articles have been published between 2011 and 2022, while only 2.7% in the last 5 years. Is there no more recent bibliography? Either the authors have not done a correct bibliographic search, or it is a topic that is no longer of much interest. Although I am not willing to accept the latter, since all the prosocial situations and the effect that empathy and other types of relationships and emotions have, are on the rise in the decision-making literature.

Response 4:  The bibliography has been updated across the whole manuscript.

Reviewer 3 Report

The topic under investigation is of scientific and social interest and is suitable for the journal Behavioral Sciences. Regarding the elaboration of the manuscript, I recommend some suggestions for improvement:

The aim of the study is not well identified in the abstract or in the introduction. I think the authors should introduce it specifically at the beginning of both sections, inclusive.

Conceptual framework: here I would recommend the argumentation of the importance of the study of this subject also for education, the promotion of prosocial behavior and education in the spirit of social and moral values; I also recommend analyzing other articles, such as:

https://link.springer.com/article/10.1023/A:1014033032440

https://www.researchgate.net/publication/339647355_The_role_of_pro-social_models_in_the_study_of_social_work_2018

Research methodology: research design is correct, well structured and cohesive; however, a clear highlighting in a special section of the results would add to the research; the authors do not specify what the possible limits of the research were; their introduction would demonstrate their honesty and frankness in experiments; In my opinion, this section would be very convenient and timely. It would also be appropriate to analyze the results from the perspective of the implications for future research in this direction.

The findings of the study should highlight the importance of the results and the need for such research for training programs for prosocial behavior in children and young people. Congratulations on the written article!

Reviewer 4 Report

This is a review of the manuscript titled "The impact of guilt and empathy on prosocial behaviour". This research consisted of two experiments that aimed to examining the effects of guilt and empathy on prosocial behavior in triadic situations. Experiment 1 showed that in a three-person situation (the participant, the victim, and the other individual), guilt drove the participants to benefited the victim at the expense of the third person without affecting their own interest. Experiment 2 showed that in a three-person situation (the participant, the victim, and the other individual), empathy drove the participants to benefited the victim at the expense of themselves but not at the expense of the third person. This manuscript is generally well written and addresses an interesting topic. It could be improved if the following concerns are addressed:

  1. In the Abstract, it is stated that, "The results show that guilt generates advantages for the victim at the expense of the transgressor and not at the expense of the third person who is not a victim." Should it be "empathy" instead of "guilt"?

  1. I suggest the authors provide a definition of prosocial behavior in the Introduction section.

  1. The authors mentioned "a three-way context" twice (p. 1 and p. 5). I am not sure what it means. Does it mean "a three-person context"?

  1. While the authors emphasize the positive interpersonal effects of guilt, I suggest the authors discuss the negative effect of guilt on personal well-being. Research has established that guilt is related to poor mental health.

  1. The authors compare guilt with shame (p. 2). It would be better to provide a brief definition of shame.

  1. It is stated that, "In our opinion, one situation in which this might occur is when the aggressor feels a simultaneous motivation both toward the victim and the third person." (p. 5) Should it be the "transgressor" instead of "aggressor"?

  1. It is stated that, "To test this hypothesis, we ran two experiments that focus upon the effects of guilt toward a third (non-victim) person present in the experimental setting." (p. 5) It should be "the effects of guilt and empathy" instead of "the effects of guilt".

  1. What does "Variable dependent" (p. 10) mean? Should it be "Dependent variable"?

  1. Post-hoc pairwise comparisons with Bonferroni correction are not necessary for the main effects of guilt and empathy. As both guilt (guilt vs. control) and empathy (communication vs. no communication) consisted of two categories, a significant main effect indicates a significant difference between the two groups.

  1. The authors may consider analyzing the data of Experiment 1 with a two-way mixed ANOVA, with guilt (guilt vs. control) as the between-subjects factor and person (the transgressor, the victim, and the third person) as the within-subjects factor.

  1. Similarly, the authors may consider analyzing the data of Experiment 2 with a three-way mixed ANOVA, with guilt (guilt vs. control) and empathy (communication vs. no- communication) as the between-subjects factors and person (the transgressor, the victim, and the third person) as the within-subjects factor.

Round 2

Reviewer 4 Report

The manuscript has been improved and is acceptable for publication in the current form.

Author Response

thanks
